# A Novel Therapeutic Reagent, KA-1002 for Alleviating Lysophosphatidic Acid-Mediated Inflammation Related Gene Expression in Swine Macrophages

**DOI:** 10.3390/ani10030534

**Published:** 2020-03-23

**Authors:** Hyeon-Jeong Hwang, Tamina Park, Miok Kim, Hee-su Shin, Wooyeon Hwang, Yong Ki Min, Suk-gil Song, Daeui Park, Chang Hoon Lee

**Affiliations:** 1Bio and Drug Discovery Division, Center for Information-Based Drug Research, Research Institute of Chemical Technology (KRICT), Daejeon 34114, Korea; hjhwang@Krict.re.kr (H.-J.H.); miok@krict.re.kr (M.K.); hsshin@krict.re.kr (H.-s.S.); wooyeon@krict.re.kr (W.H.); ykmin@krict.re.kr (Y.K.M.); 2Department of Predictive Toxicology, Korea Institute of Toxicology, Daejeon 34114, Korea; tamina.park@kitox.re.kr; 3Department of Human and Environmental Toxicology, University of Science and Technology, Daejeon 34113, Korea; 4Chungnam National University School of Medicine, Daejeon 34137, Korea; 5Department of Biochemistry, College of Natural Sciences, Chungnam National University, Daejeon 34134, Korea; 6Department of Pharmaceutical Science, Kyunghee University, Seoul 02447, Korea; 7College of Pharmacy, Chungbuk National University, Cheongju, Chungbuk 28644, Korea; songs@chungbuk.ac.kr

**Keywords:** swine macrophages, lysophosphatidic acid, antagonist, inflammation, RNA-Seq

## Abstract

**Simple Summary:**

Inflammatory diseases are a key factor reducing the productivity of animals in a livestock industrial environment. We have identified a novel lysophosphatidic acid signaling antagonist, KA-1002, which alleviates lysophosphatidic acid-mediated a broad range of inflammation related gene expression in swine macrophages. Specifically, we found that KA-1002 significantly alleviated LPA-induced genes related with inflammation such as a role of macrophages, fibroblasts and endothelial cells in rheumatoid arthritis and STAT3 signal pathway. Taken together, KA-1002 could be considered a novel therapeutic reagent candidate for swine inflammatory diseases.

**Abstract:**

Stresses and various infectious reagents caused multiple inflammatory diseases in swine in a livestock industrial environment. Therefore, there is a need for an effective therapeutic or preventive agent that could alleviate chronic and acute inflammation. We found that lysophosphatidic acid (LPA), a stress-induced potent endogenous inflammatory molecule, causes a broad range-regulation of inflammation related genes inflammation in swine macrophages. We further investigated the genome scaled transcriptional regulatory effect of a novel LPA-signaling antagonist, KA-1002 on swine macrophages, inducing the alleviated LPA-mediated inflammation related gene expression. Therefore, KA-1002 could potentially serve as a novel therapeutic or preventive agent to maintain physiologically healthy and balanced conditions of pigs.

## 1. Introduction

Various inflammatory diseases including livestock animal respiratory diseases are very common and potent destructive issues on the animal industry [1,2,3], and are caused by various environmental stresses from microbial infections, crowding, and dust [1,2,3,4]. Specifically, inflammatory respiratory diseases are critical in swine. Inflammatory respiratory diseases are highly common in modern pork production worldwide and are often referred to as porcine respiratory disease complex (PRDC) [5,6]. For those various inflammatory diseases, endogenous factors associated with those environmental stresses are also important and those inflammatory endogenous factors often weaken immunity of animals leading to susceptibility to pathogenic invasion. Among various endogenous stress-induced molecules, lysophosphatidic acid (LPA) is a powerful endogenous trigger of inflammation in a number of diseases-associated conditions [7,8,9,10]. Thus, various environmental stresses induced upregulated LPA potently triggers multiple inflammatory diseases [8,10]. LPA signaling is mediated by its receptors, LPARs and there are needs for LPA antagonists for therapeutic reagents [11,12]. In our previous report [13], we showed that LPA triggered inflammatory responses from bovine respiratory cells and bovine blood endothelial cell lines. For example, LPA strongly induced inflammatory angiogenesis and the production of inflammatory cytokine [14]. Furthermore, we found a novel LPA-antagonistic molecule, KA-1002 [13], which strongly alleviated LPA-induced inflammatory responses from bovine respiratory cells and blood endothelial cells. However, we did not understand the effect of the LPA antagonistic molecule, KA-1002 on swine cells. 

Although LPA is a well-known endogenous molecule for inflammatory diseases [15,16,17,18,19,20,21,22], the effect of LPA on swine macrophages and the molecular level mechanism for pathogenesis of LPA on swine are not fully understood until now. Thus, the investigation about the LPA mediated inflammatory condition in swine macrophages should be helpful for understanding the therapeutic mechanism of LPA-antagonistic therapeutics on porcine inflammatory diseases related abnormally upregulated LPA condition. In this study, we tried to understand LPA mediated inflammation and our previously reported LPA-antagonistic compound KA-1002, which could be used for therapeutic reagents [13] through the genome-scale transcriptional expression analysis on swine macrophages. For those goals, we investigated the genome-scale transcriptional expression pattern on swine macrophages differentially regulated by LPA treatment and the transcriptional regulation by KA-1002 on LPA induced inflammatory conditioned swine macrophages. We treated 3D4/12 cells with 100 µM LPA in the presence or absence of 20 µM KA-1002 in triplicate for 18 hours and harvested total RNAs from those cells. Using total RNAs, we tried to understand the inflammatory effect on swine macrophages and the effect of KA-1002 on LPA-treated swine macrophages through RNA-seq. Based on the RNA-Seq and gene set enrichment analysis, KA-1002 showed a regulatory effect on a broad range of inflammatory conditions such as the role of MAPK signaling, the role of IL-17, differential regulation of cytokine production in macrophages and T helper cells. Because macrophages are one of the critical regulatory immune cells equipped with multiple functions related to inflammatory diseases, those results strongly suggest that KA-1002 might be a potential therapeutic capacity on LPA signaling related swine inflammatory diseases. 

## 2. Materials and Methods

### 2.1. Drugs and Chemicals

KA-1002 was dissolved in 100% Dimethyl sulfoxide (DMSO) from Sigma-Aldrich Co Ltd (St. Louis, MO, USA, catalogue number: 276855) at a concentration of 10 mM as a stock solution stored at −20 °C and diluted in complete culture medium prior to experiments. LPA (catalogue number: L7260) was purchased from Sigma-Aldrich Co Ltd (St. Louis, MO, USA).

### 2.2. Cell Culture

3D4/21 (ATCC® CRL-2843™) swine macrophage cells were purchased from ATCC (Manassas, VA, USA) and cultured in complete Roswell Park Memorial Institute (RPMI) Dulbecco’s modified Eagle’s medium supplemented with 10% fetal bovine serum and 1% penicillin-streptomycin in a CO_2_ incubator at 95% air and 5% CO_2_, at 37 °C. For analysis of genome scaled transcription, 3D4/21 cells were treated with 100 µM LPA (Sigma-Aldrich, St. Louis, MO, USA) in the presence or absence of 20 µM KA-1002 in triplicate for 18 hours in the media and conditions described above.

### 2.3. Sample Preparation form Illumina NovaSeq Sequencing

Total RNA was isolated from cell subsets prepared as described above. An RNA sequencing library was generated using a TruSeq Stranded mRNA LT Sample Prep Kit according to the user’s instruction manual (Illumina, San Diego, CA, USA). Briefly, mRNA was separated from total RNA using oligo(dT) beads and chemically fragmented. After the double-strand cDNA synthesis of fragmented mRNA, end-repair, adenylation of the 3′ end, and sequencing adapter ligation was performed. This was followed by DNA purification with magnetic beads and PCR amplification. Finally, the amplified library was purified, quantified, and then applied for template preparation. The NovaSeq platform was utilized to generate 101-bp paired-end sequencing reads (Illumina, San Diego, CA, USA).

### 2.4. Genome Mapping and Identification of Paired-End Sequences 

All 101-bp paired-end sequence reads were mapped to the full genome sequences for *Sus scrofa domesticus* (NCBI Sscrofa11.1) using HISAT2 version 2.1.0 [23]. These mapped reads were merged for each condition (control, LPA, and LPA + KA-1002), and transfrags were assembled using StringTie version 1.3.4d [23]. These merged transfrags were quantified for each condition using the DESeq2 program. Additionally, genes with no mapped reads for each condition were excluded from the analysis. Finally, we identified DEGs that were selected by a fold change cut-off of >1.5 and an independent t-test raw *p*-value < 0.05. Finally, RNA-Seq analysis revealed 393 DEGs between control and LPA and 347 DEGs between LPA + KA-1002 and LPA. 

### 2.5. Gene Set Enrichment Test and Heat Map of Representative Gene Sets

To characterize the gene set containing differentially expressed genes (DEGs), representative gene sets were analyzed in Ingenuity Pathway Analysis (IPA, http://www.ingenuity.com). The terminologies of the gene set are listed in the IPA program. Significant tests for the mapping gene sets were performed by Fisher’s exact test (filtering options: *p*-value ≤ 0.05). The significant value of each gene set was represented as −log(*p*-value). If the *p*-value is 0.05, −log(*p*-value) is converted to 1.3. To compare the similarities in significantly expressed gene sets between LPA and LPA + KA-1002, heat map analysis was performed for the highest 14 pathways among gene sets which were filtered according to *p*-value of less than 0.05. To calculate the similarities of gene sets between two conditions, we used Euclidean distances and complete linkage as the agglomeration method for hierarchical clustering. Heat map images were generated in R with the heatmap.2 function in the gplots library (https://cran.r-project.org/web/packages/gplots/). To compare pathways between two conditions, pathways were visualized in the Differential biological pathways image. In the figure, + means upregulated pathways, and − means downregulated pathways in each condition. The value −log(*p*-value) represents the statistical significance of the selected pathway.

### 2.6. Comparative Network Analysis

To compare the expressions of significant DEGs between LPA and LPA + KA-1002, we generated a comparative network. The candidates for the network were selected as a statistical ANOVA test among control, LPA treatment, and LPA + KA-1002 treatment. To improve the reliability of the results, the statistical ANOVA test for DEGs were performed using the read counts obtained from RNA-Seq. A total of genes with a p-value less than 0.05 were screened and could make the network containing 47 genes and 65 interactions. In the networks, the interactions between DEGs were derived from resources with a combined score of greater than 900 in the STRING database. A node was represented as DEGs. Additionally, fold-changes in each gene are presented as bar charts for LPA treatment vs. control and LPA plus KA-1002 treatment vs. control. Fold changes were calculated using log2 (fold-change). 

### 2.7. Statistical Analysis

Multiple comparisons were carried out using ANOVA. Differences between two groups were evaluated using an unpaired t-test. Statistical analyses were performed using Prism software (GraphPad Inc., San Diego, CA, USA), with differences considered significant at *p* < 0.05.

## 3. Results

### 3.1. The LPA-Antagonistic Novel Compound KA-1002 Differentially Regulates the Gene Expression Profiles of LPA Treated Swine Macrophage Cells 

To investigate a genome-scale transcriptional expression pattern on swine macrophages differentially regulated by LPA and KA-1002 treatment, we analyzed gene expression profiles on a genome-wide scale of LPA-treated (LPA) and LPA plus KA-1002 (treatment) treated swine macrophage cell line, 3D4/21 compared to untreated (control) 3D4/21 cells. We treated 3D4/21 cells with 100 µM LPA in the presence or absence of 20 µM KA-1002 in triplicate for 18 hours and harvested total RNAs from those cells described in Figure 1A. As a result, we found that significant changes on genome scaled transcription in LPA treated swine macrophages compared to untreated swine macrophages. Furthermore, KA-1002 induced regulation of a broad range of gene expression in LPA plus KA-1002-treated macrophages compared to only LPA-treated macrophages. RNA-seq analysis revealed 393 DEGs between LPA and control group and 347 DEGs between the treatment group and LPA group (Figure 1B). Of the 347 DEGs between LPA plus KA-1002 treated and LPA treated cells, 246 genes were significantly upregulated, and 147 genes were significantly downregulated. Compared to LPA group, in the treatment group, 155 genes were significantly upregulated and 192 genes were significantly downregulated (Figure 1B). 

In Figure 1C, the volume represents the level of gene expression. The volume is calculated by geometric means of mapped reads between two conditions. To elucidate the differences among control, LPA, and KA-1002 treatment group, we chose the top ten upregulated and top ten downregulated genes in LPA-treated cells compared to control cells and in LPA plus KA-1002 treated cells compared to LPA treated macrophages (Table 1 and Table 2). The *NYAP2*, *TMEM35A*, *LOC110255306*, *TIAM2*, *CXCL8*, *PTGS2*, *CCL5*, *LOC110259478*, *CCL2*, and *CSF2* genes were upregulated in LPA treated swine macrophages at 7.57, 4.72, 4.69, 4.16, 3.97, 3.62, and 3.53 times, respectively, compared to untreated swine macrophages. The *LOC100515579*, *ADAP2*, *RTP4*, *NRARP*, *FAM110D*, *MGP*, *AK5*, *SLC34A1*, *KAZN*, and *SMTNL2* genes were downregulated in LPA treated macrophages at −55.60, −2.93, −2.83, −2,75, −2.65, −2.55, −2.51, −2.49, −2.38, and −2.35 times, respectively, compared to untreated swine macrophages. These genes might play an important role in LPA-treated swine macrophages. (Table 1). 

We also analyzed top ten upregulated and top ten downregulated genes in LPA plus KA-1002 treated macrophages compared to LPA treated macrophages. The Transglutaminase 3, RHO family interacting cell polarization regulator 2, Protein tyrosine phosphatase, receptor type U, Arginase 1, Gamma-aminobutyric acid type A receptor pi subunit, Family with sequence similarity 198 member B, G protein-coupled receptor 157, Leucine rich alpha-2-glycoprotein 1, Endothelial cell specific molecule 1, and Solute carrier family 4 member 4 genes were upregulated in LPA plus KA-1002 treated swine macrophages at 17.06, 7.63, 4.36, 4.00, 3.39, 2.59, 2.33, 2.25, and 2.08 times, respectively, compared to LPA treated swine macrophages. The Hyaluronan binding protein 2, Carbonic anhydrase 8, POU class 2 associating factor 1, Protein tyrosine phosphatase, receptor type R, Gametogenetin-binding protein 1-like, Cytochrome P450, family 24, subfamily A, polypeptide 1, Keratin, type I cytoskeletal 13, Leucine rich repeat containing 25, Ectonucleotide pyrophosphatase/phosphodiesterase 4 (putative), and Proline rich 15 like genes were downregulated in LPA treated macrophages at −3.15, −3.06, −2.97, −2.82, −2.62, −2.52, −2.48, −1.82, and −1.81 times, respectively, compared to LPA treated swine macrophages. These genes might play an important role in LPA-treated swine macrophages. (Table 2). 

### 3.2. Gene Set Analysis among Control, LPA-Treated, and LPA plus KA-1002 Treated Swine Macrophages

In our RNA-Seq analysis, the genomic expression of the control group and LPA group were very different. Interestingly, KA-1002 treatment group showed differential gene expression patterns from LPA and control group. The heat map for all genes with a fold change of more than +/− 2 in control, LPA and treatment group is shown in Appendix A. As shown in the heat map, LPA treated swine macrophages are clearly different from the control group in their gene expression profiles. Furthermore, the treatment group showed differential gene expression patterns from LPA treatment and control group. Based on the DEGs derived from RNA-Seq analysis, we performed gene set enrichment analysis to infer the functional differences of each group. We analyzed the 10 gene sets with the most significant p-values via the upregulated and downregulated DEGs as shown in Figure 2A. As shown in Figure 2B and Table 3, LPA group and treatment group are enriched with different sets. These results show that KA-1002 treatment in the presence of LPA could result in different biological functions compared to LPA in swine macrophages.

### 3.3. Differential Biological Pathways among Control, LPA-Treated, and LPA plus KA-1002 Treated Swine Macrophages

To determine the significant biological pathways associated with gene sets containing DEGs as a result of RNA-Seq, representative pathways were analyzed by performing Ingenuity Pathway Analysis (IPA, http://www.ingenuity.com). Gene set enrichment analysis was performed to analyze knowledge on the gene sets generated by RNA-Seq, and biological pathways were represented based on enrichment analysis with DEGs. IPA was used to search for critical pathways. IPA analysis revealed inflammation related signaling routes such as role of MAPK signaling in the pathogenesis of influenza, role of IL-17A in arthritis, IL-17 signaling, glucocorticoid receptor signaling, and differential regulation of cytokine production in macrophages from upregulated DEGs in LPA group compared to control (Figure 2A). Signaling routes of complement system, anandamide degradation, Threonine degradation II, Tec kinase signaling, and T cell receptor signaling were revealed from downregulated DEGs in LPA group compared to control (Figure 2A). Interestingly, the treatment group showed that signaling routes such as “role of MAPK signaling in the pathogenesis of influenza”, “role of IL-17A in arthritis, IL-17 signaling”, “glucocorticoid receptor signaling”, “TREM1 signaling”, “role of macrophages, fibroblasts, and endothelial cells in rheumatoid arthritis”, and “airway pathology in chronic obstructive pulmonary disease” triggered by LPA group were alleviated (Figure 2A). Furthermore, HIF-1 signaling, IL-10 signaling, leukocyte extravasation signaling from upregulated DEGs in treatment group and xenobiotic metabolism signaling, PPAR signaling, CDK5 signaling, LPS/IL-1 mediated inhibition RXR function, Ephrin A signaling from downregulated DEGs in the treatment group were shown differentially from LPA group (Figure 2A). Especially, role of macrophages pathway and role of osteoblasts pathway, granulocyte adhesion pathway, and VDR/RXR activation pathway were upregulated by LPA treatment, however, those pathways were significantly downregulated by KA-1002 treatment in macrophage (Figure 2B and Table 3). 

### 3.4. Selecting Distinguished Genes between LPA-Treated, and LPA plus KA-1002 Treated Swine Macrophages

In the comparative network with distinguished DEGs of LPA and treatment group compared to control, *CCR7*, *CCL5*, *CD44*, *DMP1*, and *SPSB2* were down-regulated by KA-1002 treatment comparing with LPA treatment condition (blue circle in Figure 3). In addition, *LCK*, *IL2RG*, *MMP9*, *MMP1*, *LRG1*, *GCNT3*, *GALNT15*, *B3GNT7*, *ADAMTS4*, *SEMA5A*, and *APOP4* were upregulated by KA-1002 treatment comparing with LPA treatment condition (red circle in Figure 3). *LCK*, *IL2RG*, and *CD44* were important for T cell immune responses. *CCL5*, *MMP1*, *GCNT3*, *SPSB2*, *DMP1*, and *LRG1* were known for their roles in innate immune cells-mediated inflammation. These transcriptome results suggest that LPA induce suppression of T cell responses and upregulation of innate immune cells-mediated inflammation. Multiple previous reports [12,13,14] support our hypothesis. Mathew D et al. showed that LPA suppresses CD8+T cell cytotoxicity function via disruption of early TCR signaling [13]. 

## 4. Discussion

A potent endogenous inflammatory molecule, LPA triggers a broad range of inflammatory responses, and has multiple synthesis routes in various environmental stress associated conditions [17,18,19,20,21,22]. Although LPA is a well-known endogenous molecule for inflammatory diseases [15,16,17,18,19,20,21,22], the effect of LPA on swine macrophages and the molecular mechanism for pathogenesis of LPA on swine are not fully understood until now. LPA is well known for its pathological functions in various inflammatory diseases, such as cancers and respiratory diseases suggesting a reasonable involvement of LPA in chronic/acute inflammatory disorders illnesses [15,16,17]. LPA effector functions are mediated by the signaling pathway through its receptors, six G-protein-coupled LPA receptors (LPARs) [11,12]. Thus LPA signaling could be inhibited using LPA antagonist such as LPARs inhibitors. Thus, LPA signaling antagonists have been attracting therapeutic targets for multiple inflammatory diseases [24]. Even though the pathological roles of LPA in livestock animals are not fully investigated in vivo study until now, many in vitro studies and several important in vivo animal models such as cows strongly suggest that LPA importantly works in multiple inflammatory diseases in livestock animals [13,16]. For example, In vivo studies of cattle for reproductive diseases might support the importance of LPA in livestock animals [25,26]. In our previous report [13], we found a novel LPA-antagonistic molecule, KA-1002, which strongly alleviated LPA-induced inflammatory responses from bovine respiratory cells and bovine blood endothelial cells. In this study, we also tried to understand the effect of the LPA antagonistic molecule, KA-1002 on swine cells with the analysis method based on transcriptome assay. 

In the previous report [27,28], downstream of LPA receptor activation, MAPK/ERK signaling pathway can be activated by LPA. In our study, we found that LPA treatment on swine macrophages strongly induced biological pathway for “role of MAPK signaling in the pathogenesis of influenza” and its related genes to be differentially regulated (Figure 3). We also identified KA-1002 treatment downregulated LPA-induced gene enrichment set of “role of MAPK signaling in the pathogenesis of influenza” (Figure 3). MAPK signaling is well-known for its role in inflammation [28]. However, the gene enrichment set of “IL-10 signaling” was increased by KA-1002 treatment compared to LPA mediated inflamed swine macrophages (Figure 3). IL-10 signaling is well known for its anti-inflammatory function [29]. Those results strongly suggest that KA-1002 alleviated LPA-specific inflammation signaling in swine macrophages and its potential as therapeutics for swine inflammatory diseases. Multiple previous reports [19,20] also support those hypotheses. Mathew D et al. showed that LPA suppresses CD8+T cell cytotoxicity function via disruption of early TCR signaling [18]. We also demonstrated that LPA mediated downregulated IL2RG and LCK were recovered by treatment of KA-1002 on swine macrophages strongly suggests that LPA suppresses TCR signaling and KA-1002 treatment recovered LPA-mediated suppression of TCR signaling (Figure 3). Those results suggest that KA-1002 might recover normal function of T cells, which were disrupted by LPA-mediated suppression of TCR signaling. Those results are also important for animal therapeutics, which could support healthy immune function of livestock animals from diseases associated immune-suppressed condition.

Taken together, these results suggest that KA-1002 treatment on LPA-induced inflamed swine macrophages caused the critical turn-off of LPA-induced inflammatory signaling. Also, our results strongly suggest KA-1002 as therapeutics supporting healthy immune function of livestock animals from various environmental stresses induced LPA associated immune suppression. As such, LPA, and LPA receptors might represent therapeutic targets for multiple inflammatory diseases of pigs, and KA-1002 could be a potential therapeutic agent as a drug or feed additive against LPA mediated multiple inflammatory diseases. It might also be used in combination with antibiotics or antiviral reagents to treat infectious diseases and their related inflammatory diseases. We did not demonstrate the therapeutic effects of KA-1002 in vivo in this study which would be a logical next step in confirming its applicability. Nonetheless, our results give a shred of strong evidence to support KA-1002 as a novel class of potential therapeutic agents for treating swine inflammatory diseases.

## 5. Conclusions

According to the results, it could be concluded that KA-1002 as a novel LPA antagonist alleviates LPA-mediated inflammatory responses by swine macrophages associated with multiple cellular signal pathways such as MAPK signaling, differential regulation of cytokine production in macrophages and T helper cells, and T cell receptor signaling. In our comparative subnetwork with DEGs, we found several critical genes altered by LPA and LPA with KA-1002 treatment (Figure 3). CCL5 level was down-regulated and CCL2, MMP1, and MMP9 were up-regulated by KA-1002 treatment suggesting that KA-1002 regulated the LPA-induced immune cell recruitment differentially (Figure 3). Those results might be supported by previous reports showing the roles of LPA in T cell recruitment and motility [19,20]. IL2RG and LCK were related to early TCR signaling. In our results, IL2RG and LCK were increased or recovered by treatment of KA-1002 from LPA-treated swine macrophages suggesting that LPA suppresses TCR signaling and KA-1002 treatment recovered LPA-mediated suppression of TCR signaling. Those results might be supported by previous reports [18]. These transcriptome results suggest that LPA induce suppression of T cell responses and upregulation of innate immune cell-mediated inflammation. Further studies using pigs are needed to confirm the therapeutic efficacy of KA-1002, which is testified for in vivo efficacy. This study is the first study to reveal that LPA-mediated multiple inflammatory signal pathways based on whole genome scaled transcription and to suggest that LPA antagonists could be a target for alleviating LPA-mediated multiple inflammatory signal pathways.

## Figures and Tables

**Figure 1 animals-10-00534-f001:**
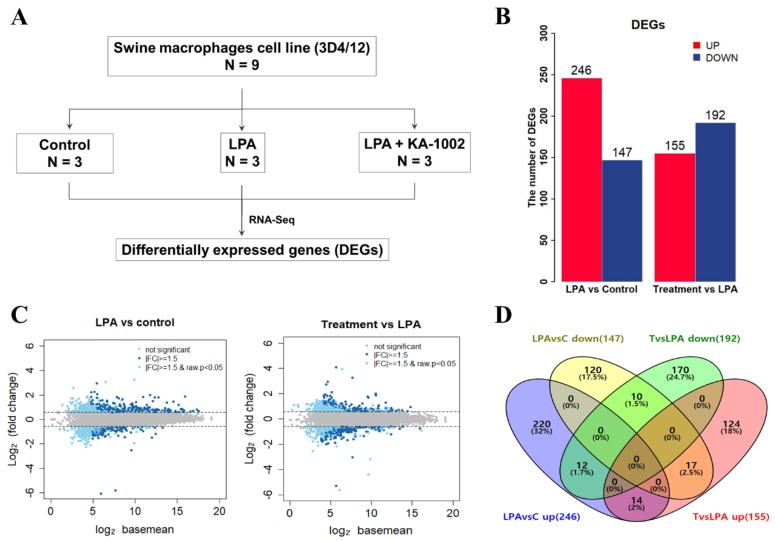
(**A**) Sample preparation for RNAseq. (**B**) Number of upregulated and downregulated differentially expressed genes (DEGs) in lysophosphatidic acid (LPA) treated swine macrophages in comparison with untreated swine macrophages (control) and DEGs in LPA + KA-1002 treated swine macrophages(treatment) in comparison with LPA treated swine macrophages (LPA). DEGs were selected by a fold change cut-off of >1.5 and *p*-value < 0.05. (**C**) Scatter dot plot indicating differentially expressed genes (DEGs) between LPA vs. control group and treatment (LPA + KA-1002) vs. LPA group. The Y axis shows fold changes in expression level (log2 value), and the X axis depicts volume. The volume indicates the level of gene expression. The volume was calculated by geometric means of mapped reads between two conditions. (**D**) Venn diagram showing relations of DEGs between LPA vs control and Treatment vs LPA group.

**Figure 2 animals-10-00534-f002:**
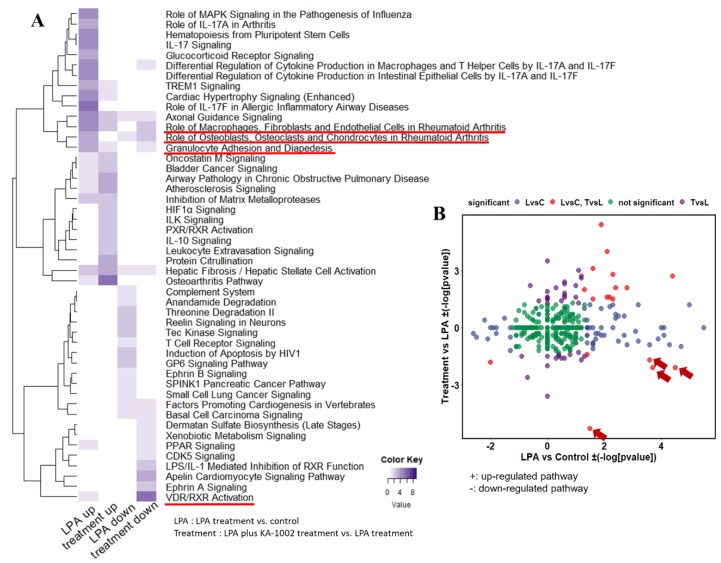
(**A**) The heat map for critical pathways contained genes with a fold change of more than +/- 2 in control, LPA-treated, and LPA plus KA-1002 treated (treatment) swine macrophages. The value of each pathway represents statistical significance as −log(*p*-value). (**B**) Differential biological pathways between LPA treated vs. control and LPA plus KA-1002 treated vs. LPA treated macrophage. + means upregulated pathways. − means downregulated pathways. The value −log(*p*-value) represents the statistical significance of the selected pathway. High-lighted four pathways were opposite regulated between two conditions. The pathways were upregulated by LPA treatment, however, significantly downregulated by KA-1002 treatment in swine macrophage.

**Figure 3 animals-10-00534-f003:**
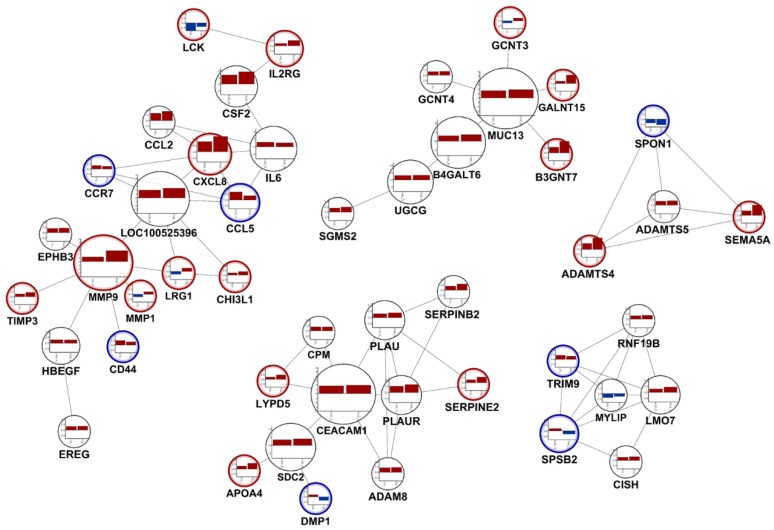
Comparative network with distinguished DEGs and their fold changes between LPA treatment and KA-1002 treatment in swine macrophages. The interactions between DEGs were derived from resources with a combined score of greater than 900 in the STRING database. A node was represented as DEGs. Additionally, fold-changes in each gene are presented as bar charts for LPA treatment vs. control (left bar) and LPA plus KA-1002 treatment vs. control (right bar). Blue circle: upregulated genes by KA-1002 treatment comparing with LPA treatment, Red circle: downregulated genes by KA-1002 treatment comparing with LPA treatment.

**Table 1 animals-10-00534-t001:** Highly differentially expressed genes in LPA treated swine macrophages vs. untreated swine macrophages.

Gene Symbol	Gene Description	Fold-Change	Adjust *p*
*NYAP2*	Neuronal tyrosine-phosphorylated phosphoinositide-3-kinase adaptor 2	7.57	2.5 × 10^−4^
*TMEM35A*	Transmembrane protein 35A	4.72	4.4 × 10^−4^
*LOC110255306*	N-acetyllactosaminide beta-1,6-N-acetylglucosaminyl-transferase-like	4.69	3.3 × 10^−3^
*TIAM2*	T-cell lymphoma invasion and metastasis 2	4.16	3.2 × 10^−3^
*CXCL8*	C-X-C motif chemokine ligand 8	3.97	1.1 × 10^−3^
*PTGS2*	Prostaglandin-endoperoxide synthase 2	3.62	5.6 × 10^−4^
*CCL5*	C-C motif chemokine ligand 5	3.53	3.8 × 10^−5^
*LOC110259478*	Uncharacterized LOC110259478	3.52	3.0 × 10^−7^
*CCL2*	Chemokine (C-C motif) ligand 2	3.47	2.6 × 10^−4^
*CSF2*	Colony stimulating factor 2	3.42	1.9 × 10^−4^
*LOC100515579*	Hydroxyacyl-thioester dehydratase type 2, mitochondrial	−55.60	1.0 × 10^−2^
*ADAP2*	ArfGAP with dual PH domains 2	−2.93	4.4 × 10^−2^
*RTP4*	Receptor transporter protein 4	−2.83	5.0 × 10^−2^
*NRARP*	NOTCH-regulated ankyrin repeat protein	−2.75	4.0 × 10^−5^
*FAM110D*	Family with sequence similarity 110 member D	−2.65	4.2 × 10^−3^
*MGP*	Matrix Gla protein	−2.55	3.0 × 10^−3^
*AK5*	Adenylate kinase 5	−2.51	2.5 × 10^−3^
*SLC34A1*	Solute carrier family 34 member 1	−2.49	1.6 × 10^−2^
*KAZN*	Kazrin, periplakin interacting protein	−2.38	3.7 × 10^−2^
*SMTNL2*	Smoothelin like 2	−2.35	8.1 × 10^−3^

**Table 2 animals-10-00534-t002:** Highly differentially expressed genes in LPA plus KA-1002 treated vs. LPA treated swine macrophages.

Gene Symbol	Gene Description	Fold-Change	Adjust *p*
*TGM3*	Transglutaminase 3	17.06	1.2 × 10^−5^
*RIPOR2*	RHO family interacting cell polarization regulator 2	7.63	2.6 × 10^−3^
*PTPRU*	Protein tyrosine phosphatase, receptor type U	4.36	2.5 × 10^−2^
*ARG1*	Arginase 1	4.00	7.8× 10^−3^
*GABRP*	Gamma-aminobutyric acid type A receptor pi subunit	3.39	3.6× 10^−11^
*FAM198B*	Family with sequence similarity 198 member B	2.59	3.3 × 10^−2^
*GPR157*	G protein-coupled receptor 157	2.33	4.5 × 10^−2^
*LRG1*	Leucine rich alpha-2-glycoprotein 1	2.32	4.2 × 10^−2^
*ESM1*	Endothelial cell specific molecule 1	2.25	9.9× 10^−3^
*SLC4A4*	Solute carrier family 4 member 4	2.08	2.5× 10^−3^
*HABP2*	Hyaluronan binding protein 2	−3.15	2.1× 10^−4^
*CA8*	Carbonic anhydrase 8	−3.06	3.7× 10^−3^
*POU2AF1*	POU class 2 associating factor 1	−2.97	2.1 × 10^−2^
*PTPRR*	Protein tyrosine phosphatase, receptor type R	−2.82	7.7× 10^−3^
*LOC100156231*	Gametogenetin-binding protein 1-like	−2.62	3.3 × 10^−2^
*CYP24A1*	Cytochrome P450, family 24, subfamily A, polypeptide 1	−2.60	5.0× 10^−5^
*LOC100515166*	Keratin, type I cytoskeletal 13	−2.52	2.5× 10^−3^
*LRRC25*	Leucine rich repeat containing 25	−2.48	3.6 × 10^−2^
*ENPP4*	Ectonucleotide pyrophosphatase/phosphodiesterase 4	−1.82	3.6 × 10^−2^
*PRR15L*	Proline rich 15 like	−1.81	2.0 × 10^−3^

**Table 3 animals-10-00534-t003:** Differential biological pathways between LPA treated vs. control and LPA plus KA-1002 treated vs. LPA treated swine macrophage.

Pathway	−log(*p*-value)		Number of Gene	Related Genes
	LPA vs. Control (Upregulated)	Treatment vs. LPA (Downregulated)		
Role of Macrophages, Fibroblasts and Endothelial Cells in Rheumatoid Arthritis	4.5	−2.1	21	*HNF1A*, *RRAS2*, *ADAMTS4*, *DKK3*, *IL18R1*, *SOCS3*, *WNT9A*, *CXCL8*, *IL1RL1*, *CCL2*, *TRAF1*, *NGFR*, *PRKCE*, *MMP1*, *CSF2*, *IL6*, *CCL5*, *PGF*, *IL17RC*, *VEGFC*, *WNT11*
Role of Osteoblasts, Osteoclasts and Chondrocytes in Rheumatoid Arthritis	3.7	−2.1	15	*BMP7*, *DKK3*, *IL18R1*, *WNT9A*, *IL11*, *HNF1A*, *BCL2*, *MMP1*, *ADAMTS5*, *WNT11*, *IL1RL1*, *CSF2*, *IL6*, *NGFR*, *ADAMTS4*
Granulocyte Adhesion and Diapedesis	3.6	−1.7	13	*CCL5*, *CLDN7*, *SDC2*, *CLDN6*, *CXCL8*, *MMP1*, *ITGA1*, *MMP7*, *IL1RL1*, *CCL2*, *NGFR*, *PECAM1*, *MMP9*
VDR/RXR Activation	1.5	−5.3	10	*HES1*, *CCL5*, *HR*, *PRKCE*, *CYP24A1*, *COL13A1*, *KLF4*, *IL1RL1*, *MED1*, *CSF2*
STAT3 Pathway	1.4	−1.4	11	*IL18R1*, *SOCS3*, *IL2RG*, *TGFBR3*, *CISH*, *IL17RC*, *BCL2*, *IL1RL1*, *RRAS2*, *NGFR*, *IL10RA*

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
