# Peer review of "A Novel Therapeutic Reagent, KA-1002 for Alleviating Lysophosphatidic Acid-Mediated Inflammation Related Gene Expression in Swine Macrophages"

_animals, 2020, doi:10.3390/ani10030534_

Round 1

Reviewer 1 Report

The manuscript entitled "A novel therapeutic reagent, KA-1002 for alleviating lysophosphatidic acid-mediated inflammation related gene expression in swine macrophages by Hyeon-Jeong Hwang et al. describes an interesting study on novel possibly therapeutic reagent KA-1002. Authors have found that KA-1002 has a capacity to alleviate lysophosphatidic acid-mediated inflammation related gene expression in swine macrophages. Results of the study indicate that KA-1002 could be considered a novel therapeutic reagent candidate for swine inflammatory diseases.

In opinion of the referee this is well designed and propely conducted study of possibly high practical value.

Although the manuscript is well written the referee noticed some lingual problems that need to be corrected:

  1. Introduction. -   Line 48: ...on the animal industry. [1-3], and is (dot?, should be are) caused by...
  2. Results. -   Line 254: ...treatment group showed that almost (almost ???) signaling routes triggered by LPA...
  3. Discussion. -   Line 295: ...is the main substrate of autotoxin which present (present ???) is present in circulation...

In opinion of the referee this is an interesting and well written manuscript. Therefore referee recommends this manuscript for publication in journal "Animals".

Author Response

The manuscript entitled "A novel therapeutic reagent, KA-1002 for alleviating lysophosphatidic acid-mediated inflammation related gene expression in swine macrophages by Hyeon-Jeong Hwang et al. describes an interesting study on novel possibly therapeutic reagent KA-1002. Authors have found that KA-1002 has a capacity to alleviate lysophosphatidic acid-mediated inflammation related gene expression in swine macrophages. Results of the study indicate that KA-1002 could be considered a novel therapeutic reagent candidate for swine inflammatory diseases.

In opinion of the referee this is well designed and propely conducted study of possibly high practical value.

Although the manuscript is well written the referee noticed some lingual problems that need to be corrected:

  1.  Line 48: ...on the animal industry. [1-3], and is (dot?, should be are) caused by...

   Answer : I removed dot in the sentence.

  1.  Line 254: ...treatment group showed that almost (almost ???) signaling routes triggered by LPA...

   A: I revised the sentences more clearly, in line 260-264.

  1. -   Line 295: ...is the main substrate of autotoxin which present (present ???) is present in circulation...

   Answer: another reviewer recommended that those sentences is not fit in the whole     story, thus I delete those sentence.

In opinion of the referee this is an interesting and well written manuscript. Therefore referee recommends this manuscript for publication in journal "Animals".

Answer: Thanks for your wonderful comment.

Reviewer 2 Report

RUNNING TITLE

Line 28-29: it must be shorter 

SIMPLE SUMMARY

It needs to be expanded 

INTRODUCTION

As a whole the introduction is somewhat poor and unclear. It should certainly be expanded in all its parts, starting from the animal model and why respiratory diseases in pigs are so serious in the livestock economy, indicate in more detail what LPA is and what it does, why use Ka-1002.

Only by detailing the state of the art the purpose of the proposed study can be better understood.

Add for example the below article to expand the introduction:

  • Front Cell Neurosci.2019 Nov 29;13:531. doi: 10.3389/fncel.2019.00531. eCollection 2019. Small-Molecule Lysophosphatidic Acid Receptor 5 (LPAR5) Antagonists: Versatile Pharmacological Tools to Regulate Inflammatory Signaling in BV-2 Microglia Cells. Plastira I1, Joshi L1, Bernhart E1, Schoene J2, Specker E2, Nazare M2,3, Sattler W1,4.
  • BiochimBiophys Acta Mol Cell Biol Lipids.2020 May;1865(5):158641. doi: 10.1016/j.bbalip.2020.158641. Epub 2020 Jan 29. The roles of autotaxin/lysophosphatidic acid in immune regulation and asthma. Kim SJ1, Moon HG1, Park GY2.
  • BiochimBiophys Acta.2013 Jan;1831(1):86-92. doi: 10.1016/j.bbalip.2012.06.014. Epub 2012 Jul 15. Lysophosphatidic acid (LPA) and its receptors: role in airway inflammation and remodeling. Zhao Y1, Natarajan V.
  • Mediators Inflamm.2017;2017:9173090. doi: 10.1155/2017/9173090. Epub 2017 Dec 21. Autotaxin-Lysophosphatidic Acid: From Inflammation to Cancer Development.Valdés-Rives SA1, González-Arenas A1.
  • Anim Health Res Rev.2011 Dec;12(2):133-48. doi: 10.1017/S1466252311000120. Polymicrobial respiratory disease in pigs.Opriessnig T1, Giménez-Lirola LG, Halbur PG.

6)     Porcine Respiratory Disease Complex https://www.ncbi.nlm.nih.gov/books/NBK2481/?report=reader

Line 61-70: “As a result …. Inflammatory disease” This lines are  not part of the introduction but they are results, not surprisingly the thought begins with as a result. consequently it must be integrated into the discussion

MATHERIAL AND METHODS

Fr each reagent used please add also the number of catalogue

Line 76: indicate the final concentration used for KA-1002 and Ki-16425

RESULTS

Line 143-151 This lines describe the goal of the total work and must be moved to the final part of the introduction, when after the aim of the work you briefly describe what you have done

Line 261-263 in these lines you are specifying a explanation for your results, this part must therefore be integrated into the discussion as it is not an objective result

Line 275-278 in these lines you are specifying a explanation for your results, this part must therefore be integrated into the discussion as it is not an objective result

DISCUSSION

Line 293-296 It is not clear why at this point you cited autotoxin and lysophosphatidylcholine. In my opinion it is better that you start the discussion taking back LPA functions and expanded with its pahway

Line 303: I am not convinced that humans can be included among model animals. Please rewrite the sentenceLine 303 and 305: please explain and expanded what you have put in quotes

Line 327-337 These lines must be integrated in the conclusions

REFERENCES

Correct the references following the guidelines of the journal:

-        after the name of an author, before the surname of the following author you have to put semicolon and not comma

-       Write all authors name

Author Response

RUNNING TITLE

Q: Line 28-29: it must be shorter

A: Yes, we revised running title in line 28-29 on page 1.

SIMPLE SUMMARY

Q: It needs to be expanded

A: Yes, we added summarized results in simple summary in line 29 – 36 on page 1.

INTRODUCTION

Q: As a whole the introduction is somewhat poor and unclear. It should certainly be expanded in all its parts, starting from the animal model and why respiratory diseases in pigs are so serious in the livestock economy, indicate in more detail what LPA is and what it does, why use Ka-1002. Only by detailing the state of the art the purpose of the proposed study can be better understood.

A: We revised the introduction with more information related our research.

Q: Add for example the below article to expand the introduction:Front Cell Neurosci.2019 Nov 29;13:531. doi: 10.3389/fncel.2019.00531. eCollection 2019. Small-Molecule Lysophosphatidic Acid Receptor 5 (LPAR5) Antagonists: Versatile Pharmacological Tools to Regulate Inflammatory Signaling in BV-2 Microglia Cells. Plastira I1, Joshi L1, Bernhart E1, Schoene J2, Specker E2, Nazare M2,3, Sattler W1,4. BiochimBiophys Acta Mol Cell Biol Lipids.2020 May;1865(5):158641. doi: 10.1016/j.bbalip.2020.158641. Epub 2020 Jan 29. The roles of autotaxin/lysophosphatidic acid in immune regulation and asthma. Kim SJ1, Moon HG1, Park GY2. BiochimBiophys Acta.2013 Jan;1831(1):86-92. doi: 10.1016/j.bbalip.2012.06.014. Epub 2012 Jul 15. Lysophosphatidic acid (LPA) and its receptors: role in airway inflammation and remodeling. Zhao Y1, Natarajan V. Mediators Inflamm.2017;2017:9173090. doi: 10.1155/2017/9173090. Epub 2017 Dec 21. Autotaxin-Lysophosphatidic Acid: From Inflammation to Cancer Development.Valdés-Rives SA1, González-Arenas A1. Anim Health Res Rev.2011 Dec;12(2):133-48. doi: 10.1017/S1466252311000120. Polymicrobial respiratory disease in pigs.Opriessnig T1, Giménez-Lirola LG, Halbur PG. 6)     Porcine Respiratory Disease Complex https://www.ncbi.nlm.nih.gov/books/NBK2481/?report=reader

A: Yes, we added recommended references in our manuscript.

Q: Line 61-70: “As a result …. Inflammatory disease” This lines are not part of the introduction but they are results, not surprisingly the thought begins with as a result. consequently it must be integrated into the discussion

A: Yes, we moved “. As a result, we found that significant changes on genome scaled transcription in LPA treated swine macrophages compared to untreated swine macrophages. Furthermore, KA-1002 induced regulation of a broad range of gene expression in LPA plus KA-1002-treated macrophages compared to only LPA-treated macrophages.” into result section in line 152-155 on page 4. In our discussion, we discussed the detailed results.

MATHERIAL AND METHODS

Q: Fr each reagent used please add also the number of catalogue

A: We added catalogue numbers

Q: Line 76: indicate the final concentration used for KA-1002 and Ki-16425

A: Ki-16425 was not used in this experiment, it was my mistake. We delete the sentence.

Round 2

Reviewer 2 Report

Dear  authors your work has an undoubted scientific value but I would kindly suggest that you give equal importance in drafting the article, pay more attention and be less rushed

The introduction is still poor and unclear. The addition of two sentences (lines 51-54 and lines 58-59) is not enough to frame the state of the art well in order also to make the aim of the work clear to the readers. In the haste to respond to required revisions, be careful of spelling mistakes, as in line 53 offen.

RESULTS

Line 277-280 in these lines you are specifying a explanation for your results, this part must therefore be integrated into the discussion as it is not an objective result

DISCUSSION

Line 310 and 313: please explain and expanded what you have put in quotes

Line 329-338 These lines must be delete at this point and integrated in the conclusions

Author Response

Dear  authors your work has an undoubted scientific value but I would kindly suggest that you give equal importance in drafting the article, pay more attention and be less rushed

 Q: The introduction is still poor and unclear. The addition of two sentences (lines 51-54 and lines 58-59) is not enough to frame the state of the art well in order also to make the aim of the work clear to the readers. In the haste to respond to required revisions, be careful of spelling mistakes, as in line 53 offen.

A: Thanks for your comment. I revised offen into often. I also did spell-check and added more information in introduction.

RESULTS

Q: Line 277-280 in these lines you are specifying a explanation for your results, this part must therefore be integrated into the discussion as it is not an objective result

A: We moved those sentences into discussion. 

DISCUSSION

Q: Line 310 and 313: please explain and expanded what you have put in quotes

A: We added more information in those lines.

Q: Line 329-338 These lines must be delete at this point and integrated in the conclusions

A: We moved those sentences into conclusion.  Thanks for your nice comments.

Round 3

Reviewer 2 Report

line 75 change goal with goals

line 114 there is the reference written in full. change according authors instructions

line 326 change In the previous report with In previous reports